# Reconfigurable Terahertz Spatial Deflection Varifocal Metamirror

**DOI:** 10.3390/mi14071313

**Published:** 2023-06-27

**Authors:** Jianhui Fang, Renbin Zhong, Boli Xu, Huimin Zhang, Qian Wu, Benzheng Guo, Jianian Wang, Zhenhua Wu, Min Hu, Kaichun Zhang, Diwei Liu

**Affiliations:** Terahertz Research Center, School of Electronic Science and Engineering, Cooperative Innovation Centre of Terahertz Science, University of Electronic Science and Technology of China, Chengdu 610054, China

**Keywords:** metamirror, metal–graphene metasurface, digitally encoded, focal deflection

## Abstract

A traditional optical lens usually has a fixed focus, and its focus controlling relies on a bulky lens component, which makes integration difficult. In this study, we propose a kind of terahertz spatial varifocal metamirror with a consistent metal–graphene unit structure whose focus can be flexibly adjusted. The focus deflection angle can be theoretically defined by superimposing certain encoded sequence on it according to Fourier convolution theorem. The configurable metamirror allows for the deflection of the focus position in space. The proposed configuration approach presents a design concept and offers potential advancements in the field of developing novel terahertz devices.

## 1. Introduction

Metasurfaces, composed of two-dimensional structures with units arranged periodically at subwavelength scales [1], offer flexible and efficient control over the phase [2,3,4], amplitude [5,6], polarization [7,8], and other characteristics of electromagnetic waves. Of these capabilities, phase modulation has been widely adopted in diverse fields, including wavefront control [9,10,11,12,13], holographic imaging [14,15,16,17], polarization manipulation [18,19], and electromagnetic stealth [20,21].

In the initial stages of research, upon finalizing the design of a metasurface, its electromagnetic properties and response characteristics are determined [22]. Subsequently, researchers have explored the utilization of graphene [23,24,25,26], liquid crystals [27,28], and phase change materials [29,30,31] in the design of adjustable metasurfaces. Of these materials, graphene stands out as a two-dimensional material composed of honeycomb carbon atoms, exhibiting tunability through chemical doping or electrostatic biasing [23]. Takumi et al. presented a tunable metasurface comprising a one-dimensional array of graphene ribbons, allowing for phase change by manipulating the Fermi levels, thereby enabling complete control over the wavefront [32]. Muhammad et al. proposed a design strategy for a three-functional metasurface, utilizing a single anisotropic nanoresonator combined with geometric phase-based spin-decoupling and Malus’s law intensity modulation. The design enables independent encoding of three different pieces of information and yields distinct responses under various incident polarization scenarios [33]. Regarding encoded metasurfaces, Cui et al. proposed the concept of a digitally programmable metasurface, employing two discrete units with “0” and “π” phases to emulate digital “0” and “1” states [34]. This approach enables adjustments to be performed using digital control technology, providing precise control and manipulation of the metasurface’s properties.

The traditional optical lens is bulky and difficult to integrate and relies on surface curvature at various radial locations to determine the focal position [35]. In recent years, varifocal metalenses have gained significant attention in miniaturized integrated photonic devices. Several approaches have been proposed to achieve focus control along the optical axis, including stretchable substrates, microelectromechanical systems, and thermo-optic effects [36]. However, these methods often encounter challenges such as mechanical stretching and complex structures [37]. In contrast, encoded metalenses enable flexible and intricate wavefront manipulation by independently encoding each unit, making them highly promising for the development of ultra-thin varifocal metalenses [38]. Nevertheless, a key challenge remains: adjusting the focal position in space rather than solely along the optical axis.

This study presents a novel approach to deflecting the focus position spatially on the reconfigurable digitally encoded metasurface, which consists of uniform metal–graphene unit structures. Phase modulation is achieved by adjusting the Fermi levels of the graphene strips. By incorporating Fourier convolution, the metasurface allows for flexible adjustments of the deflection angle and focal position in space. Additionally, a specific strategy is introduced to enable finer manipulation of the focal point.

## 2. Structure Design

The requirement of achieving multi-bit encoded metasurfaces with phase differences of 180° (1-bit), 90° (2-bit), or 45° (3-bit) between the encoded units is necessary [28]. In this work, a 2-bit metal-graphene metasurface is designed for wavefront modulation, as shown in Figure 1a, featuring a consistent unit structure with each unit possessing a specific phase distribution enabled by the reconfigurable properties of graphene. Figure 1b illustrates the unit structure of the elliptical split-ring resonator (ESRR), composed of two elliptical split metal rings separated by a graphene strip. The major axis (a), minor axis (b), and width (w) of the elliptical split ring are 20 µm, 14 µm, and 2 µm, respectively. The graphene strip has a width of $w_{x}$ = 6 µm and a length of $w_{y}$ = 32 µm. The metal–graphene structure is tiled on a reflective gold substrate, with a TOPAS dielectric layer separating them, as depicted in Figure 1c. Both the gold substrate and ESRR have a thickness of $t_{1}$ = 1 µm, the TOPAS dielectric layer has a thickness of $t_{2}$ = 9 µm, characterized by a dielectric constant of 2.35, and the electrical conductivity of gold is 4.561 × 10^7^ S/m. In the terahertz and infrared frequency ranges, the electrical conductivity of graphene is primarily attributed to intraband transitions, which can be mathematically expressed by the Kubo formula [25]:(1)$$\sigma \left(\omega \right)=\frac{ie^{2}E_{F}}{\pi \hbar^{2}\left(\omega +i\tau^{-1}\right)},$$
where ω is the radian frequency, e is the charge of the electron, ħ is reduced Planck constant, $k_{B}$ is Boltzmann’s constant, 𝑇 is room temperature, and the electron relaxation time is τ=1 ps. The Fermi level can be expressed as $E_{F}=\hbar v_{F}\sqrt{\pi n}$, where $v_{F}$ = 10^6^ m/s represents the Fermi velocity and n denotes the carrier density. The proposed metasurface allows for phase control of individual units by altering the surface conductivity of graphene through the modulation of the Fermi levels. A linearly polarized plane wave is incident on the metasurface along the -z direction. Using a finite integration method, we simulated the reflection amplitude, phase spectrum, and electric field distribution of the designed metasurface with periodic boundary conditions set in the x and y directions, and an open boundary condition in the z direction. 

In order to construct a 2-bit encoded metasurface, we designed four highly reflective encoded units with a continuous phase difference of π/2 between each unit. These units were labeled as “0”, “1”, “2”, and “3”, corresponding to their reflection phases of 0°, 90°, 180°, and 270°, respectively. The reflection phase gradients were achieved by carefully setting the Fermi levels of the graphene strips within each unit. Figure 2a,b illustrate the reflection amplitude and phase distributions, respectively, of the four encoded graphene units at different Fermi levels. The phase exhibits continuous variation within the frequency range of 1.8–2 THz, with a coverage range of up to 270°. By selecting specific groups of Fermi levels, we can obtain the desired continuous phase differences at corresponding frequency points. For instance, at 1.86 THz, the phase distribution of the 2-bit encoded metasurface with reflection phases of 0°, 90°, 180°, and 270° can be achieved by applying Fermi levels of 0.28 eV, 0.38 eV, 0.52 eV, and 1.15 eV to the graphene strips of the four encoded units, respectively.

According to the generalized Snell’s law [4], when a y-polarized plane wave is incident on the proposed metasurface at the angle $\theta_{i}$, the direction of the reflected beam will deviate from the vertical direction by the angle $\theta_{r}$:(2)$$sin\theta_{r}-sin\theta_{i}=\frac{\lambda_{0}}{2\pi n_{i}}\frac{d\Phi \left(x\right)}{dx},$$
where $\theta_{r}$ and $\theta_{i}$ denote the reflected and incident angles of the electromagnetic waves, respectively. $\lambda_{0}$ represents the vacuum wavelength, and $n_{i}$ denotes the refractive index of the surrounding medium. Moreover,$\;d\Phi \left(x\right)$ and dx represent the phase difference and geometric distance between adjacent units, respectively. In the case of two-dimensional (2D) systems, the modulation of the scattered wavefront direction can be achieved using the following formulas:(3)$$\theta_{r}=\sin^{-1}\left(\lambda \sqrt{\frac{1}{L_{x}{}^{2}}+\frac{1}{L_{y}{}^{2}}}\right)=\sqrt{{\left(\sin\theta_{x}\right)}^{2}+{\left(\sin\theta_{y}\right)}^{2}},$$
(4)$$\varphi_{r}=\tan^{-1}\left(\frac{d\Phi \left(y\right)⁄dy\;}{d\Phi \left(x\right)⁄dx\;}\right),$$
where $\theta_{r}$ and $\varphi_{r}$ refer to the pitching and azimuth angles of the reflected wavefront, which are determined by the metasurface phase gradients of $d\Phi \left(x\right)⁄dx$ and $d\Phi \left(y\right)⁄dy$, respectively. 

## 3. Results and Discussion

It is possible to realize a planar metasurface focusing mirror by utilizing metasurface units to construct an equivalent phase distribution of a traditional optical lens. Drawing inspiration from the phase calculation formula of a traditional optical lens, the ideal compensatory phase required by each unit of the metasurface can be mathematically expressed as [39]
(5)$$\Phi \left(x,y\right)=\frac{2\pi }{\lambda }\left(\sqrt{x^{2}+y^{2}+F^{2}}-F\right),$$
where F represents the focal length of the metasurface, and the term $x^{2}+y^{2}$ denotes the squared distance from a point $\left(x,y\right)$ to the center of the metamirror.

The actual discrete approximation of the ideal compensating phase distributions is realized by employing four encoded units to construct the planar focusing metamirror. Different Fermi levels are applied to the graphene strips of different ESRR units to obtain metamirrors of different focal lengths. Figure 3a showcases a centrally focused metamirror with a focal length of 1mm encoded by four units, each unit labeled with the corresponding reflection phases. At a frequency of 1.86 THz, the focusing of the metamirror was simulated by setting open boundary conditions in the x, y, and z directions. Figure 3b,c depict the electric field distribution on the y-z plane at X = 0 mm and the x-y plane at Z = 1 mm, respectively. It is evident that the reflected plane wave converges at the focus point (0, 0, 1) mm along the *Z*-axis. Clearly, the constructed coplanar varifocal metamirror enables fine control of the phase distributions of incident electromagnetic waves and facilitates adjustment of the focus point in the Z-direction, resulting in a significant reduction of the volume of optical instruments.

Furthermore, the focus position of the aforementioned metasurface can be flexibly deflected along the X- or *Y*-axis based on the principle of convolution [40]. As shown in Figure 4, by applying encoded sequences Sx2 (0°-0°-90°-90°-180°-180°-270°-270°...) in the X direction, which corresponds to reconstructing the metamirror by superimposing the designed Sx2 gradient sequence onto the initial encoded metasurface, the focus position is modified. Figure 4b demonstrates a 25° shift of the reflected electric field focus along the *X*-axis. According to the electromagnetic field theory, the electric field distribution on the metasurface and far-field scattering patterns of the reconstructed metamirror follow the Fourier transform, and the encoded sequences can be predesigned using Fourier convolution. Similarly, in Figure 5a, when the Sx3(0°-0°-0°-90°-90°-90°-180°-180°-180°-270°-270°-270°...) encoded sequences are superimposed on the same original metamirror, the focus position is shifted by 16° along the *X*-axis (Figure 5b). According to Equation (2), the reflection angles of the two reconstructed metamirrors can be theoretically calculated as $\theta_{rs2}=24.05{}^{\circ}$ and $\theta_{rs3}=15.76{}^{\circ}$, respectively. The consistency between the simulated and calculated results indicates that the focus position of the presented encoded metamirror can be accurately controlled by superimposing encoded sequences onto it. The complex hybrid encoded sequences obtained by multiple superimposing encoded gradient sequences enable more precise control of the focal point deflection. The range of focal point deflection in the one-dimensional direction is 3.8°–53°. Taking the encoded sequences Sx2 and Sy3 as examples, the range of focal point deflection in the one-dimensional direction is 3.8°–53°, and the encoded matrix is represented as follows:(6)$$\mathrm{Sx}2={\left[\begin{array}{c} \begin{array}{cc} \begin{array}{ccc} 0 & 0 & 1\\ 0 & 0 & 1 \end{array} & \begin{array}{cc} \begin{array}{ccc} 1 & 2 & 2\\ 1 & 2 & 2 \end{array} & \begin{array}{ccc} 3 & 3 & \cdots \\ 3 & 3 & \cdots \end{array} \end{array}\\ \vdots & \vdots \end{array}\\ \begin{array}{cc} \begin{array}{ccc} 0 & 0 & 1\\ 0 & 0 & 1 \end{array} & \begin{array}{cc} \begin{array}{ccc} 1 & 2 & 2\\ 1 & 2 & 2 \end{array} & \begin{array}{ccc} 3 & 3 & \cdots \\ 3 & 3 & \cdots \end{array} \end{array} \end{array} \end{array}\right]}_{25\times 25}\mathrm{Sy}3={\left[\begin{array}{ccc} \begin{array}{cc} 0 & 0\\ 0 & 0\\ 0 & 0 \end{array} & \cdots & \begin{array}{cc} 0 & 0\\ 0 & 0\\ 0 & 0 \end{array}\\ \begin{array}{c} \begin{array}{cc} 1 & 1\\ 1 & 1\\ 1 & 1 \end{array}\\ \begin{array}{cc} 2 & 2\\ 2 & 2\\ 2 & 2 \end{array}\\ \begin{array}{c} \begin{array}{cc} 3 & 3\\ 3 & 3\\ 3 & 3 \end{array}\\ \begin{array}{cc} \vdots & \vdots \end{array} \end{array} \end{array} & \cdots & \begin{array}{c} \begin{array}{cc} 1 & 1\\ 1 & 1\\ 1 & 1 \end{array}\\ \begin{array}{cc} 2 & 2\\ 2 & 2\\ 2 & 2 \end{array}\\ \begin{array}{c} \begin{array}{cc} 3 & 3\\ 3 & 3\\ 3 & 3 \end{array}\\ \begin{array}{cc} \vdots & \vdots \end{array} \end{array} \end{array} \end{array}\right]}_{25\times 25},$$


By superimposing encoded sequences in the Y direction, the focus can also be shifted along the *Y*-axis direction, as demonstrated in Figure 6. Figure 6a,b present the schematic diagrams of the reconstructed metamirror superimposed with encoded sequences Sy2 and Sy3, respectively. The corresponding reflected electric field distributions of the reconstructed metamirror are shown in Figure 6c,d. As anticipated, these distributions indicate that the focus has deviated from the *Z*-axis by 25° and 16° in the *Y*-axis direction, respectively.

When the metamirror is simultaneously superimposed with encoded sequences in the X and Y directions, a two-dimensional or even spatial shift in the focus can be achieved. Figure 7 illustrates the process of superimposing encoded sequences Sy2 and Sx3 on the original focusing metamirror. The resulting electric field distributions are presented in Figure 8a,b, revealing that the focus is shifted to the corner in the X–Y direction. Consequently, the coordinate of the focus point moves from (0, 0, 1) mm to (0.27, 0.38, 0.88) mm. The pitch and azimuth angles can be calculated as 28° and 54°, respectively, based on Equations (3) and (4). Furthermore, Figure 8c,d demonstrate that the focus is deflected by 22° in the Y–Z plane and 12° in the X–Z plane simultaneously.

In general, by superimposing pre-designed encoded sequences, the focus of the metamirror can be adjusted in space with certain focal lengths. The deflection angle of the focus can be theoretically calculated by the equation $\theta_{r}=\sin^{-1}\left(\sin\theta_{rs1}\pm \sin\theta_{rs2}\right)$, where $\theta_{rs1}$ and $\theta_{rs2}$ represent the reflection angles corresponding to the two superimposed encoded sequences, respectively. It is worth noting that finer spatial focusing deflection can be achieved by performing multiple superpositions of encoded sequences on the metamirror.

## 4. Conclusions

In conclusion, this study has successfully constructed a reconfigurable encoded spatial focus deflection metamirror. The metamirror features a consistent metal–graphene unit structure and allows for modulation of the phase distribution by adjusting the Fermi levels of the graphene strip in different units. At a frequency of 1.86 THz, the focus of the metamirror can be independently adjusted by superimposing encoded sequences along either the X or Y direction. Moreover, by simultaneously superimposing encoded sequences along both the X and Y directions, finer spatial control over the focus position was achieved. The obtained results align well with the principle of Fourier convolution. The proposed reconfigurable hybrid metasurface holds great potential for the design of innovative optical lenses, dynamic holographic displays, and laser beam steering.

## Figures and Tables

**Figure 1 micromachines-14-01313-f001:**
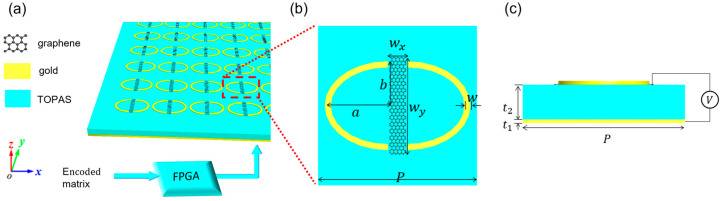
Proposed metal –graphene metasurface: (**a**) 3D view; (**b**) schematic of the top layer, *p* = 50 µm; a = 20 µm; b = 14 µm; $w_{x}$ = 6 µm; $w_{y}$ = 32 µm; (**c**) side view of the unit, for which the thicknesses of the gold substrate and TOPAS spacer are denoted as $t_{1}$ = 1 µm; $t_{2}$ = 9 µm.

**Figure 2 micromachines-14-01313-f002:**
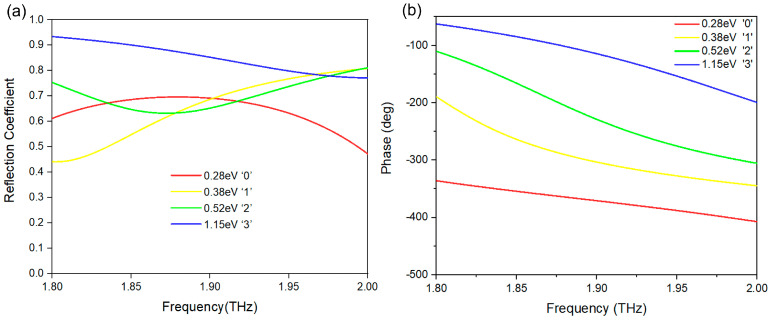
Simulated reflection amplitude (**a**) and phase spectra (**b**) of four different graphene Fermi levels encoded units of “0”, “1”, “2”, and “3”.

**Figure 3 micromachines-14-01313-f003:**
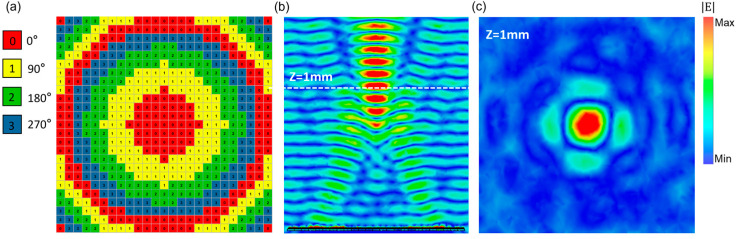
(**a**) Schematic diagram of encoded metamirror: (**b**) electric field distribution in the X = 0 mm plane; (**c**) electric field distribution in the Z = 1 mm plane.

**Figure 4 micromachines-14-01313-f004:**
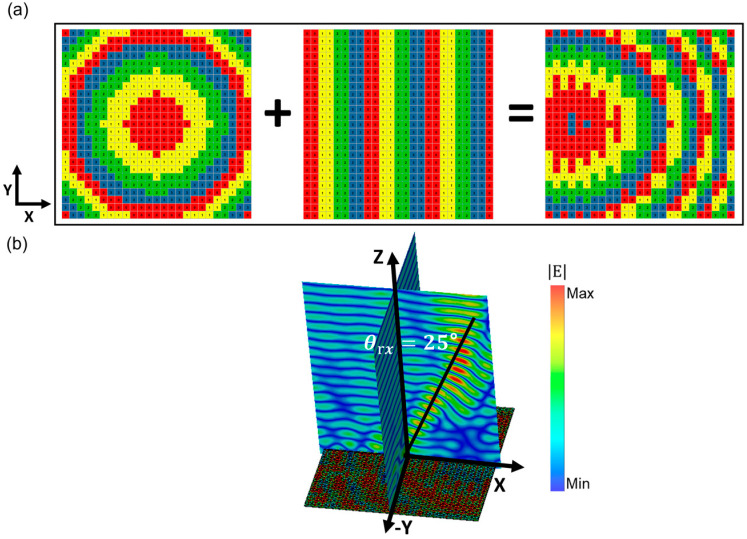
Schematic diagram of the metamirror constructed by superimposed encoded sequences of Sx2 (**a**) and corresponding electric field distributions (**b**).

**Figure 5 micromachines-14-01313-f005:**
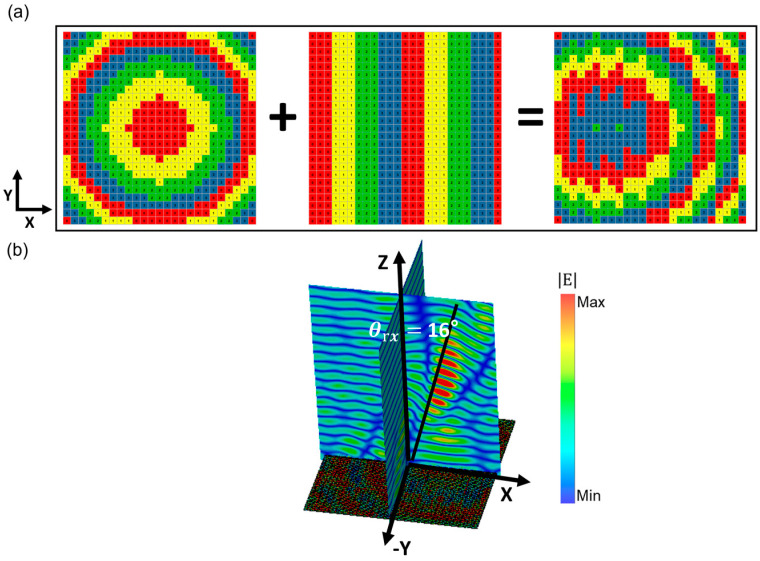
Schematic diagram of the metamirror constructed by superimposed encoded sequences of Sx3 (**a**) and corresponding electric field distributions (**b**).

**Figure 6 micromachines-14-01313-f006:**
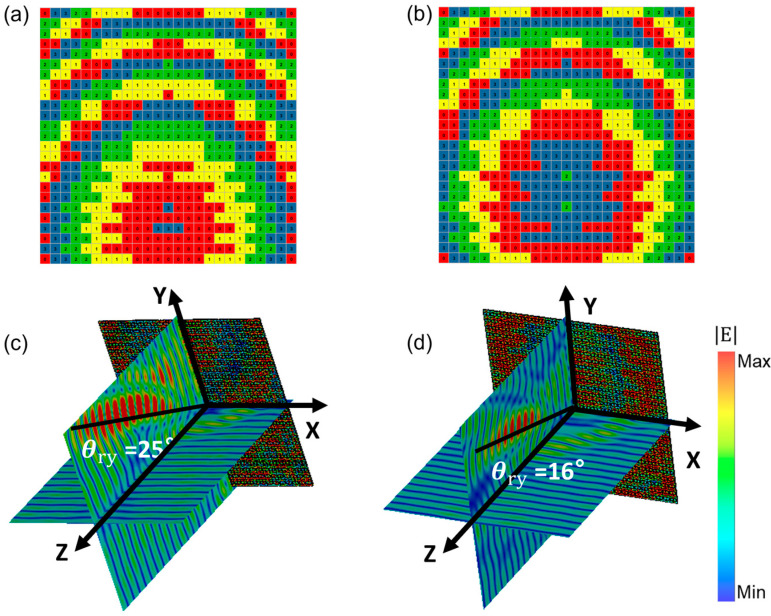
Metamirror schematic with superimposed encoded sequences (**a**) Sy2; (**b**) Sy3, and corresponding electric field distributions (**c**) Sy2; (**d**) Sy3.

**Figure 7 micromachines-14-01313-f007:**
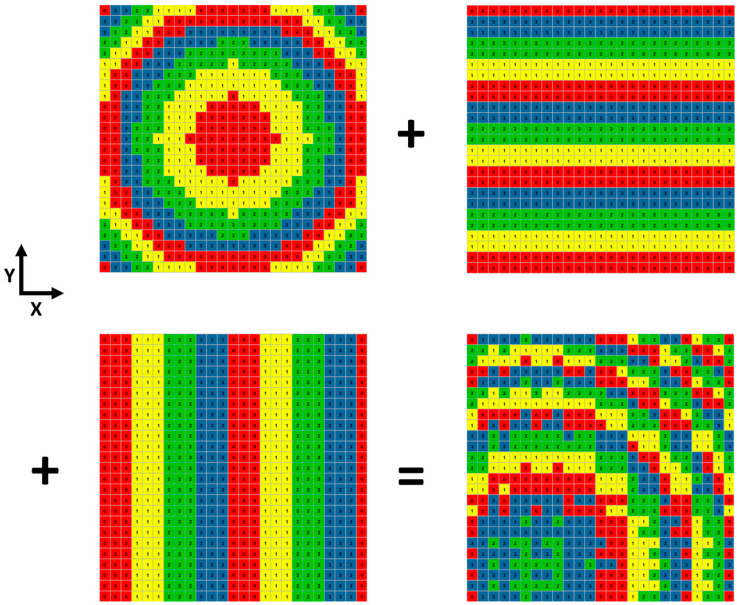
Schematic diagram of the 2D encoded metamirror, with Sy2 and Sx3 encoded sequences, superimposed simultaneously.

**Figure 8 micromachines-14-01313-f008:**
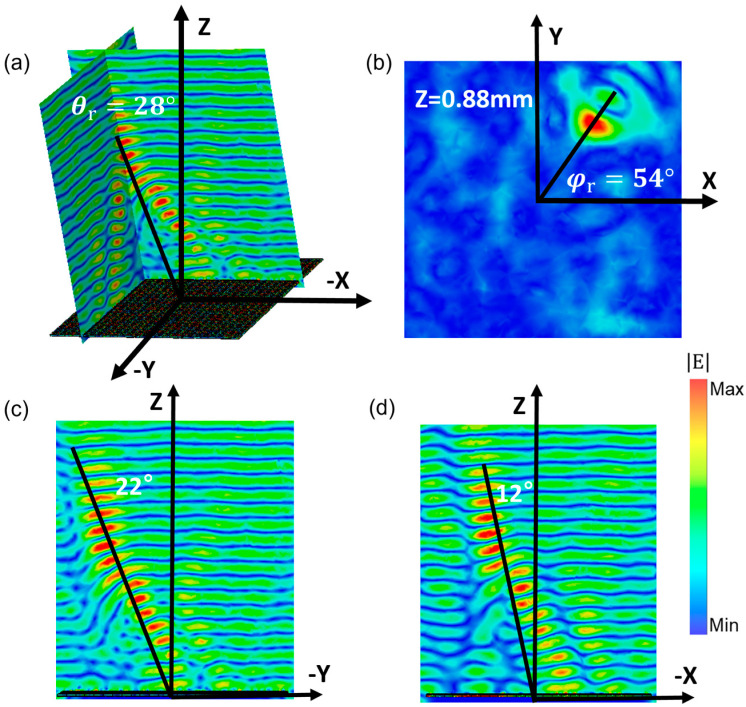
Electric field of the encoded metamirror: (**a**) pitch angle; (**b**) azimuth angle; and the corresponding electric fields on (**c**) Y–Z plane; (**d**) X–Z plane.

## Data Availability

Data available on request from the authors.

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
