# Peer review of "Reconfigurable Terahertz Spatial Deflection Varifocal Metamirror"

_micromachines, 2023, doi:10.3390/mi14071313_

Round 1

Reviewer 1 Report (Previous Reviewer 2)

The author proposed a metasurface composed of metal and graphene with consistent unit structures that can be reconfigured digitally. The graphene strip can modulate the phase in real-time by adjusting the Fermi levels. Combined with Fourier convolution, it can flexibly adjust the deflection angle of focus in space and focus position. Additionally, the study proposes a specific strategy to control spatial focus position precisely. However, similar works have been published [Acs Photonics 7, no. 6 (2020): 1425-1435; Carbon 149 (2019): 125-138.] Therefore, I recommend that the manuscript be accepted after undergoing minor revisions. My comments are detailed below:

1.      Apart from using graphene to manipulate Fermi Levels, what is the novelty of this work? In addition, the structure of the metamaterial of the split-ring resonator (SRR) seems common. Furthermore, there are no experimental results demonstrated in this work. Therefore, the validity of the numerical results may make quite doubtful. Could the author give some discussion about the question above?

2.      If Figure 1(a)-(c) is deemed possible to be fabricated, how does the author tune the Fermi energy if no metal gate or pad is deposited on top?

3.      The equivalent circuit model should be established to characterize metamaterial design. Furthermore, the equivalent metasurface layer’s impedance should be discussed to characterize the impedance of the substrate and free space.

4.      This manuscript should present the simulation of far-field distributions of the scattering pattern of the metamaterial.

5.      To help the readers have a more comprehensive understanding of the new research on metamaterials, I suggest supplementing some latest works about Folding metamaterials with extremely strong electromagnetic resonance [Photonics Research 10, no. 9 (2022): 2215-2222.]; intelligent coding metasurface holograms by physics-assisted unsupervised generative adversarial network [Photonics Research 9, no. 4 (2021): B159-B167]; Dual-band multifunctional coding metasurface with a mingled anisotropic aperture for polarized manipulation in full space [Photonics Research 10, no. 2 (2022): 416-425]; Polarization-vortex holographic encryption based on photo-oxidation of a plasmonic disk [Optics Letters 47, no. 16 (2022): 4127-4130].

Author Response

Response to Reviewer 1 Comments

Point 1: Apart from using graphene to manipulate Fermi Levels, what is the novelty of this work? In addition, the structure of the metamaterial of the split-ring resonator (SRR) seems common. Furthermore, there are no experimental results demonstrated in this work. Therefore, the validity of the numerical results may make quite doubtful. Could the author give some discussion about the question above?

Response 1: Please provide your response for Point 1. (in red)

         Thank you to the reviewer for their suggestions and comments on the manuscript. The manipulation of Fermi levels in graphene and the metamaterial structure of split-ring resonator (SRRs) is indeed common. However, the intention of our manuscript is to demonstrate an approach to control the spatial deflection of the focal point of a metamirror but not the feature of unit structure itself. That is combination of Fourier convolution theorem with the superposition of encoded sequences on a metasurface to achieve spatial deflection of the focal point and focus scanning near the metasurface. It provides an alternative design approach on reconfigurable terahertz devices for diverse applications, including the design of innovative optical lenses, dynamic holographic displays, and laser beam steering.

The numerical results presented in our manuscript is simulated by software of CST, it is a well known and commonly used commercial electromagnetic simulation software. Although experimental validation has not been conducted at this stage due to the limited time, but many previous research works have demonstrated that the experimental results of metamaterial structures designed by CST are highly coincide with the simulated ones. For example, Experimental Demonstration of >230° Phase Modulation in Gate-Tunable Graphene–Gold Reconfigurable Mid-Infrared Metasurfaces [Nano Lett. 2017, 17, 5, 3027–3034]; Inductive Tuning of Fano-Resonant Metasurfaces Using Plasmonic Response of Graphene in the Mid-Infrared [Nano Lett. 2013, 13, 3, 1111–1117]; Graphene-Integrated Reconfigurable Metasurface for Independent Manipulation of Reflection Magnitude and Phase [Adv Opt Mater 2021, 9, 2001950].

Point 2: If Figure 1(a)-(c) is deemed possible to be fabricated, how does the author tune the Fermi energy if no metal gate or pad is deposited on top?

Response 2: Please provide your response for Point 2. (in red)

We appreciate your question very much, FPGA can be used to store the designed phase distribution, and the graphene in each unit are programmed by FPGA enabling the spatial deflection of the focal position. The schematic diagram illustrating the loading of FPGA and graphene bias voltages has been added to Figure 1a and 1c in the main text. In the works of other researchers, we can observe the integration of metasurface units with PIN diodes. By controlling the diode switches through FPGA, it is easy to achieve variations in the phase of the unit structure, enabling real-time switching between different encodings of the metasurface. The numerical simulation analysis and experimental verification results presented are highly consistent [1-3]. Similarly, experimental validation has demonstrated the feasibility of modulating graphene conductivity by applying a modulation voltage between the metasurface and graphene thin film [4,5]. But independent control of patterned graphene has only been theoretically explained in some relevant papers [6]. In our manuscript, a tunable Graphene metasurface is proposed theoretically and demonstrated numerically.

  1. Sun Y L, Zhang X G, Yu Q, et al. Infrared-controlled programmable metasurface. Science Bulletin, 2020, 65(11): 883-888.
  2. Yang H, Cao X, Yang F, et al. A programmable metasurface with dynamic polarization, scattering and focusing control. Scientific reports, 2016, 6(1): 1-11.
  3. Zhang N, Chen K, Zheng Y, et al. Programmable coding metasurface for dual-band independent real-time beam control. IEEE Journal on Emerging and Selected Topics in Circuits and Systems, 2020, 10(1): 20-28.
  4. Xia L, Zou Y, Zhang M, et al. Multi-mode graphene based terahertz amplitude modulation enhanced by hollow cross H-structured metasurface. Physica Scripta, 2019, 94(12): 125701.
  5. Feng J L, Wu L S, Mao J F. Switchable broadband/narrowband absorber based on a hybrid metasurface of graphene and metal structures. Optics Express, 2023, 31(8): 12220-12231.
  6. Hosseininejad S E, Rouhi K, Neshat M, et al. Reprogrammable graphene-based metasurface mirror with adaptive focal point for THz imaging. Scientific reports, 2019, 9(1): 1-9.

Point 3: The equivalent circuit model should be established to characterize metamaterial design. Furthermore, the equivalent metasurface layer’s impedance should be discussed to characterize the impedance of the substrate and free space.

Response 3: Please provide your response for Point 3. (in red)

The analysis of metamaterial characteristics involves multiple approaches, with the equivalent circuit model being acknowledged for its effectiveness in representing metamaterial properties. However, when it comes to the design of phased metasurfaces, the reflected phase responses of individual metasurface units are not entirely identical. Consequently, employing an equivalent circuit model fails to accurately capture the electromagnetic characteristics. Therefore, this article refrains from utilizing this approach for analysis. Instead, we adopt the Generalized Snell's law to comprehensively analyze the electromagnetic properties of the metasurface as a whole, enabling us to overcome the limitations associated with individual unit responses.

Point 4: This manuscript should present the simulation of far-field distributions of the scattering pattern of the metamaterial.

Response 4: Please provide your response for Point 4. (in red)

Thank you for pointing out this issue. There is indeed an incorrect description in the "near-field region" part of the content. Regarding the far-field distribution of the scattering patterns of the metasurface, I would like to provide the following explanation: In our manuscript, the designed reflector is a device that focuses the electromagnetic waves with the focal point near the metasurface. It manipulates the reflected waves to be focused near the metasurface, resulting in a reflected focused electric field distribution. Therefore, our paper does not analyze or discuss the far-field distribution results of the metasurface.

Point 5: To help the readers have a more comprehensive understanding of the new research on metamaterials, I suggest supplementing some latest works about Folding metamaterials with extremely strong electromagnetic resonance [Photonics Research 10, no. 9 (2022): 2215-2222.]; intelligent coding metasurface holograms by physics-assisted unsupervised generative adversarial network [Photonics Research 9, no. 4 (2021): B159-B167]; Dual-band multifunctional coding metasurface with a mingled anisotropic aperture for polarized manipulation in full space [Photonics Research 10, no. 2 (2022): 416-425]; Active beam manipulation and convolution operation in VO2-integrated coding terahertz metasurfaces [Optics Letters 47, no. 2 (2022): 441-444]; Polarization-vortex holographic encryption based on photo-oxidation of a plasmonic disk [Optics Letters 47, no. 16 (2022): 4127-4130].

Response 5: Please provide your response for Point 5. (in red)

We appreciate the reviewer's recommendation and the suggested references have been added into the references list of this manuscript, e.g. 26,31 in References. By the way, some other recent research achievements in the field of metamaterials also have been introduced, such as intelligent coding metasurfaces, multifunctional coding metasurfaces, active beam manipulation, and holographic encryption, e.g. 13,17,26,31,33 in References.

Reviewer 2 Report (New Reviewer)

Fang et al. proposed a terahertz varifocal metasurface enabling flexible and adjustable focus deflection. The focus deflection angle is defined by imposing certain encoded sequences according to the Fourier convolution theorem. Overall quality of the work is satisfactory, however, before acceptance, I think authors should make minor changes. My comments/questions for the authors are listed below:

1)      Keeping in mind the fabrication of the device how Fermi levels of the graphene strips within each unit can be achieved/tuned for achieving the required reflection phase gradients?

2)      For making the introduction more comprehensive I would suggest authors should discuss these tunable metasurfaces in the introduction section doi.org/10.1002/advs.202203962; doi.org/10.1007/s11468-018-0789-0

3)      Please provide more details regarding the simulations. Used Software and numerical calculation settings?

4)      Please provide more background commentary for Figure 3a. How this phase distribution is calculated?

Minor grammar changes are required.

Author Response

Response to Reviewer 2 Comments

Point 1: Keeping in mind the fabrication of the device how Fermi levels of the graphene strips within each unit can be achieved/tuned for achieving the required reflection phase gradients?

Response 1: Please provide your response for Point 1. (in red)

Thank you for your attention to this issue. Each unit of the designed metasurface is independent. By utilizing FPGA, we independently apply bias voltages to the metasurface units at different positions. From previous work by other researchers, we can also learn that switching control of the PIN diode can achieve phase changes in the unit structure, and using FPGA can quickly apply different encoded matrices to the metasurface to achieve reflected phase gradients. And the numerical simulation analysis and experimental verification results are consisted [1-3]. For Graphene, the Fermi levels is modulated by changing the bias voltage according to the formula . The units with different Fermi levels exhibit varying reflected phase responses. Some relevant papers have also demonstrated the feasibility of applying a modulation voltage between the metasurface and the graphene film to modulate the conductivity of graphene [4,5]. By arranging the units with phase gradients according to the predetermined layout, an encoded matrix is formed, which enabling the spatial deflection of the focal position. Through FPGA control, the desired phase gradient metasurface is designed.

  1. Sun Y L, Zhang X G, Yu Q, et al. Infrared-controlled programmable metasurface. Science Bulletin, 2020, 65(11): 883-888.
  2. Yang H, Cao X, Yang F, et al. A programmable metasurface with dynamic polarization, scattering and focusing control. Scientific reports, 2016, 6(1): 1-11.
  3. Zhang N, Chen K, Zheng Y, et al. Programmable coding metasurface for dual-band independent real-time beam control. IEEE Journal on Emerging and Selected Topics in Circuits and Systems, 2020, 10(1): 20-28.
  4. Xia L, Zou Y, Zhang M, et al. Multi-mode graphene based terahertz amplitude modulation enhanced by hollow cross H-structured metasurface. Physica Scripta, 2019, 94(12): 125701.
  5. Feng J L, Wu L S, Mao J F. Switchable broadband/narrowband absorber based on a hybrid metasurface of graphene and metal structures. Optics Express, 2023, 31(8): 12220-12231.

Point 2: For making the introduction more comprehensive I would suggest authors should discuss these tunable metasurfaces in the introduction section doi.org/10.1002/advs.202203962; doi.org/10.1007/s11468-018-0789-0

Response 2: Please provide your response for Point 2. (in red)

Thank you to the reviewers for their feedback. The design of an adjustable metasurface has been supplemented in the introduction section on page 1, line 32. The relevant references have been added into the references list of this manuscript, e.g. 33 in References.

Point 3: Please provide more details regarding the simulations. Used Software and numerical calculation settings?

Response 3: Please provide your response for Point 3. (in red)

Thank you for your suggestions on software and numerical calculation settings. A detailed description of the simulation setup of the metamirror is added to the manuscript on page 4, line 133. The revised text is ”the focusing of the metamirror was simulated by setting open boundary conditions in the x, y, and z directions.”

Point 4: Please provide more background commentary for Figure 3a. How this phase distribution is calculated?

Response 4: Please provide your response for Point 4. (in red)

I appreciate the reviewer for your questions regarding the calculation of phase distribution. In the manuscript, we manily presented a reconfigurable digitally encoded metasurface consist of uniform metal-graphene unit structures, and phase modulation is achieved by adjusting the Fermi levels of the graphene strips in different units. It means that an equivalent phase distribution of the traditional optical lens can be constructed on a planar metasurface, so to realize focusing metamirror. Drawing inspiration from the phase calculation formula for traditional optical lens, the ideal compensating phase for each metasurface unit is expressed as ,  and the actual discrete approximation of the ideal compensating phase distribution are achieved by employing four encoding units.  The detailed content has be supplemented on page 4, line 119 of the manuscript.

This manuscript is a resubmission of an earlier submission. The following is a list of the peer review reports and author responses from that submission.

Round 1

Reviewer 1 Report

In the manuscript titled "Terahertz spatial varifocal metalens with reconfigurable digitally encoding metal-graphene terahertz metasurface," the authors present simulation results for a tunable meta-lens operating in the terahertz band. They use a metal-graphene unit structure to dynamically tune the phase based on the Fermi level of the graphene strip, allowing the meta-lens to control the focus at various locations and achieve real-time modulation of the terahertz wavefront.

 However, this approach has already been proposed in a paper published in 2020 ("Graphene-enabled reconfigurable terahertz wavefront modulator based on complete Fermi level modulated phase," New Journal of Physics 22.6 (2020): 063054). In addition to designing tunable meta-lens, the 2020 paper also includes a dynamic Airy beam generator and back reflector, all employing Fermi-level modulation.

 Therefore, as there is no new contribution and a lack of innovation,  The current manuscript is unsuitable to be published in Micromachines.

Reviewer 2 Report

The author proposed a metasurface composed of metal and graphene with consistent unit structures that can be reconfigured digitally. The graphene strip can modulate the phase in real-time by adjusting the Fermi levels. Combined with Fourier convolution, it can flexibly adjust the deflection angle of focus in space and focus position. Additionally, the study proposes a specific strategy to control spatial focus position precisely. However, similar works have been published [Acs Photonics 7, no. 6 (2020): 1425-1435; Carbon 149 (2019): 125-138.] Therefore, I recommend that the manuscript be accepted after undergoing major revisions. My comments are detailed below:

1.      Apart from using graphene to manipulate Fermi Levels, what is the novelty of this work? In addition, the structure of the metamaterial of the split-ring resonator (SRR) seems common. Furthermore, there are no experimental results demonstrated in this work. Therefore, the validity of the numerical results may make quite doubtful. Could the author give some discussion about the question above?

2.      If Figure 1(a)-(c) is deemed possible to be fabricated, how does the author tune the Fermi energy if no metal gate or pad is deposited on top?

3.      How did the author input the Fermi levels into the simulation, then what is the setup or boundary condition of the simulation? The author should explain the detail of this.

4.      The description of color to the 2-bit encoding number should be put as an inset of the figure of the schematic diagram of encoding metalens. For example, red represents “0”, yellow represents “1”, and soon. In addition, the label of the encoding number is not clear due to the image’s low resolution.

5.      How did the author define the superimposed encoding sequences? Could the author explain it using the transformation of the 2x2 matrix?

6.      How much is the maximum angle of the focus deflection that can be achieved using the proposed metamaterial? Could the author put some discussion of previous work about this to improve the discussion section?

7.      To help the readers have a more comprehensive understanding of the new research on metamaterials, I suggest supplementing some latest works about Folding metamaterials with extremely strong electromagnetic resonance [Photonics Research 10, no. 9 (2022): 2215-2222.]; intelligent coding metasurface holograms by physics-assisted unsupervised generative adversarial network [Photonics Research 9, no. 4 (2021): B159-B167]; Dual-band multifunctional coding metasurface with a mingled anisotropic aperture for polarized manipulation in full space [Photonics Research 10, no. 2 (2022): 416-425]; Active beam manipulation and convolution operation in VO 2-integrated coding terahertz metasurfaces [Optics Letters 47, no. 2 (2022): 441-444]; Polarization-vortex holographic encryption based on photo-oxidation of a plasmonic disk [Optics Letters 47, no. 16 (2022): 4127-4130].